# Release of nanodiscs from charged nano-droplets in the electrospray ionization revealed by molecular dynamics simulations

Beibei Wang [1✉] & D. Peter Tieleman [2✉]

Electrospray ionization (ESI) is essential for application of mass spectrometry in biological systems, as it prevents the analyte being split into fragments. However, due to lack of a clear understanding of the mechanism of ESI, the interpretation of mass spectra is often ambiguous. This is a particular challenge for complex biological systems. Here, we focus on systems that include nanodiscs as membrane environment, which are essential for membrane proteins. We performed microsecond atomistic molecular dynamics simulations to study the release of nanodiscs from highly charged nano-droplets into the gas phase, the late stage of ESI. We observed two distinct major scenarios, highlighting the diversity of morphologies of gaseous product ions. Our simulations are in reasonable agreement with experimental results. Our work provides a detailed atomistic view of the ESI process of a heterogeneous system (lipid nanodisc), which may give insights into the interpretation of mass spectra of all lipid-protein systems.

[1] Centre for Advanced Materials Research, Beijing Normal University, Zhuhai 519087, People's Republic of China. [2] Department of Biological Sciences and Centre for Molecular Simulation, University of Calgary, Calgary T2N 1N4, Canada. ✉email: bbwang@bnu.edu.cn; tieleman@ucalgary.ca

Electrospray ionization (ESI) is especially useful in producing ions for macromolecules, because it can prevent these molecules splitting into fragments[1,2]. In the ESI process, droplets of analyte solution are emitted from the Taylor cone into a heated gas environment. These droplets are positively charged due to the presence of excess $H^+$, $NH_4^+$, or $Na^+$. After a cascade of solvent evaporation and jet fission, droplets split into off-spring nano-droplets with at most one analyte ion, and ultimately the analyte ions are released into the gas phase. The evaporated neutral solvents are carried away by the heated carrier gas, and the analyte ions get into the mass spectrometer, and are sorted by the mass-to-charge ($m/z$) ratio.

Three models have been proposed for the late stage of the ESI process, release of the analyte ions from nano-droplets[3]. The ion evaporation model (IEM)[4] proposed that the electrostatic field at the nano-droplet surface triggers analyte ejection. According to the charged residue model (CRM)[5], the solvent of nanodroplets evaporates, leaving the analyte in the gas phase. In the third model, the chain ejection model (CEM)[6], the analyte gets extruded from the nano-droplet surface step by step. The CEM model can be considered a special case of the IEM model. It has been proposed, but with some controversy, that the three models are applicable for different molecules: IEM for small ions, CRM for large globular analytes and CEM for disordered peptides/proteins and polymers[3,7]. It is challenging to describe the late stage of the ESI process in detail experimentally[8], because nano-droplets are small and short-lived with a life-time of microseconds (μs)[9]. Therefore, theoretical approaches have been used to extract the relation between charge state and the mass/shape of the gaseous ions from the available mass spectrometry results[10–13]. Computational methods, especially molecular dynamics (MD) simulations[14,15], have also been performed to investigate the charge-induced conformational transitions of proteins[16–18] and polymers[19] in vacuum, the relation of charge states versus nano-droplet morphologies[20], and release of analyte ions, such as small ions[21–23], peptides[24], polymers[25–27], global proteins[28], and protein complexes[29], from charged nano-droplets[30]. Here, we explored the spatio-temporal evolution of charged nano-droplets with empty (without protein) lipid nanodiscs at atomistic resolution using MD simulations.

Nanodiscs have become a very powerful tool for a variety of biochemical and biophysical techniques, such as mass spectrometry (MS)[31,32], nuclear magnetic resonance (NMR)[33,34], and cryo-electron microscopy[35,36]. Nanodiscs consist of a lipid bilayer surrounded by amphipathic helical belt proteins, termed membrane scaffold proteins (MSPs)[37,38]. Two MSP monomers form an antiparallel dimer[39]. Nanodiscs are soluble in water, and provide a native-like lipid bilayer environment for membrane proteins, allowing to analyze the surrounding lipid environment of membrane proteins. Nanodiscs tend to be more stable than micelles and vesicles, and the size is easy to control by changing the sequence length of MSPs, yielding monodisperse particles of designed size and composition[40].

Native MS of intact nanodiscs was reported for nanodiscs containing the scaffold protein MSP1D1 and DMPC or POPC lipids[41–43]. At low $m/z$, several sharp peaks arise from isolated lipid monomers, dimers, and trimers. At high $m/z$, the spectra show two primary features. Several consecutive broad peaks were observed, with a series of peaks with a fine spacing superimposed over each broad peak, which was due to different number of lipids in the nanodiscs, indicating partial disassembly during the ESI process. At low collision energies, the nanodisc was largely intact, while at higher collision energies, the nanodisc collapses significantly. When a membrane protein was embedded in the nanodisc, at low collision energies, the membrane protein with lipid in contact could be released. At higher collision energies, the complex dissociated into its components.

Similar mass spectra were observed from MS analysis of heterogeneous nanodiscs as well[43]. The complexity of these peaks at high $m/z$ presents a challenge for interpretation and assignment of the spectra. It also indicates that the behavior of nanodiscs in the ESI process may not be as simple as that for experiments in bulk solution. Here, we use MD simulations to investigate two key questions: what is the behavior of nanodiscs in the ESI process and to what extent do nanodiscs maintain their structures after the ESI process. Two distinct major scenarios were observed, and resulted in diverse morphologies of gaseous product ions.

## Results

We constructed a structural model of a nanodisc containing two MSP1 proteins, 186 DMPC lipids, 81 sodium ions and ~75,000 water molecules, yielding a system of more than 250,000 atoms with a net 69 positive charge. The radius of this nanodroplet is about 8 nm, which to our knowledge is the largest droplet studied by microsecond atomistic MD simulations to date. After equilibration, we investigated the release of nanodiscs from positively charged droplets. The temperature of the late stage of the ESI process is still controversial; some studies have demonstrated it is at room temperature[44], while some studies supported higher temperatures[28,45]. Therefore, here ten simulations with 6.6 μs total simulation time were produced at different temperatures: 300 K (labeled 300-1, 300-2, 300-3 and 300-4), 370 K (labeled 370-1, 370-2 and 370-3) and 370 → 450 K (370 K for the initial 75 ns, and 450 K for the last 75 ns, and labeled 450-1, 450-2 and 450-3) (Table S1). In the simulations at higher temperature, the water evaporation is much faster, and the nanodisc undergoes more significant collapse (Figs. 1–3 and S2–10), which is consistent with the mass spectrometry results at higher collision energies[31]. However, the observed ESI process at different temperatures is consistent in general.

**The ESI processes.** The initially cubic simulation system forms a spherical droplet quickly within hundreds of picoseconds after the ESI simulations start, to reduce the surface tension. Solvent evaporation gradually reduces the droplet size, accompanied by the occasional ejection of solvated $Na^+$ (Figs. 1, 2, and 3a). Some of the ions migrate to the droplet surface, resulting in droplet distortion and the formation of spike-like protrusion. Some of the ions bind to the surface of the nanodisc. These observations are common in all ten simulations, but the behavior of the nanodisc revealed two distinct processes, which we termed at-center and off-center scenarios. The at-center scenario conforms to the CRM model, while the off-center scenario is more consistent with a CRM/CEM hybrid model.

The at-center scenario is dominant, followed by seven of the ten simulations (300-2, 300-3, 300-4, 370-2, 370-3, 450-2 and 450-3 (Figs. 1 and S3–S8)). Nanodiscs exhibit consistent behavior in these seven simulations. Taking a closer look at the 300-3 simulation (Fig. 1), the nanodisc stays within the nano-droplet interior over almost 1000 ns (Fig. S2). As the remaining solvent molecules are not enough to cover the surface of the nanodisc, the bilayer starts to collapse (Fig. 1f). Hydrophobic tails of DMPC lipids at the water-air interface turn into the vacuum phase and gradually cover the surface of the remaining water molecules, to reduce the surface tension. Meanwhile, to reduce the electrostatic strength, the detachment of two lipid-ion complexes, $[1DMPC + 1Na + 8H2O]^{1+}$ and $[5DMPC + 2Na + 38H2O]^{2+}$, was observed at high temperature in the 450-3 simulation (Fig. S8). In this process, the two MSP monomers maintain their dimer conformations, but with different degrees of dissociation. After about 1200 ns, 710 water molecules are still bound to the nanodisc, and are hard to evaporate. The product gaseous ions of

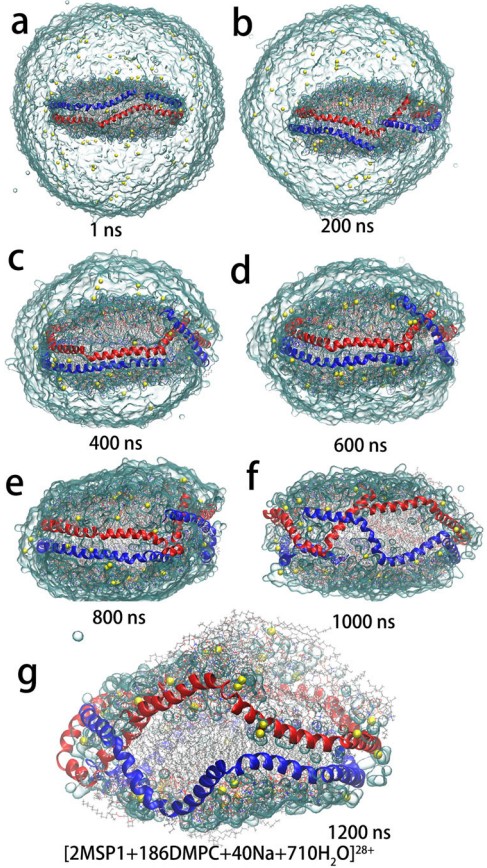

**Fig. 1 Evolution of the charged nanodisc droplet in the at-center scenario.** **a–g** The snapshots at different simulation times in the 300-3 simulation. The initial system contains a membrane scaffold protein 1 (MSP1) dimer (with -12 charges) with two monomers colored in blue and red, respectively, 186 dimyristoylphosphatidylcholine (DMPC) lipids, and 81 sodium ions (yellow spheres). The water surface is shown as light blue glass bubble. The MSPs are displayed in cartoon representation.

the 300-3 simulation have the composition of $[2MSP1 + 186 DMPC + 40Na + 710H_2O]^{28+}$ (Table 1).

In the off-center simulations (300-1 shown in Figs. 2, 370–2 in Figs. S9, and 450–2 in Fig. S10), the process displays a dramatically different behavior of the expulsion of one MSP monomer into the gas phase. Taking the 300-1 simulation, for example, the nanodisc gradually migrates to the surface of the droplet starting at about 50 ns. After about 400 ns, the nanodisc edge touches the water/air interface and the MSP1 dimer starts to dissociate at this site. This leaves the hydrophobic lipid core exposed to the vacuum and leads to lipids escaping from this site, forming a monolayer on the droplet surface. Meanwhile, the C-terminal of one MSP monomer is extruded into the vacuum (Fig. 2e), and detached from the droplet step by step as the simulation progresses until about 1000 ns; then the expelled part adopts a more compact conformation at low temperature than those at high temperatures (Figs. 2 and S9–S11). The protruded chain enables charge migration from the droplet, and loads with 10/17/13 Na$^+$ in the 300-1, 370-2, and 450-2 simulations respectively. The charged and polar sites of the expelled protein remain solvated and protected by small water and lipid clusters to avoid exposure to the gas phase (Figs. 2 and S9–10). Meanwhile, the escaped lipids gradually cover the droplet surface, finally forming a reverse micelle (Fig. 2h) and enveloping thousands of remaining water molecules. In principle, lipid-containing clusters could be expelled from the droplet as solvent evaporates. This was

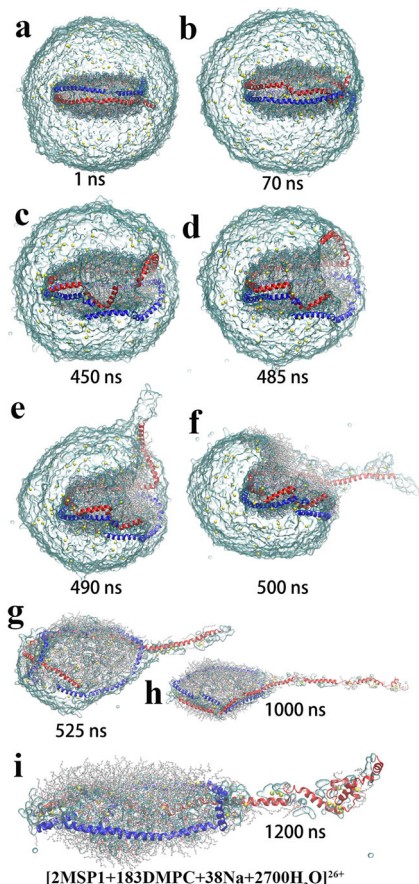

**Fig. 2 Evolution of the charged nanodisc droplet in the off-center scenario. a–i** The snapshots at different simulation times in the 300-1 simulation. The representation is the same as in Fig. 1.

observed in simulations 300-1 and 450-2 (Table 1). Three lipid molecules detach from the C-terminal of the expelled MSP monomer without net charges due to the weak binding in the 300-1 simulation, and in the 450-2 simulation, one off-spring ion, $[12DMPC + 3Na + 4H_2O]^{3+}$, is released from the droplet. Finally, the 300-1 simulation yields a product gaseous ion with the composition of $[2MSP1 + 183DMPC + 38Na + 2700H_2O]^{26+}$, with an extended shape.

**The shapes of product gaseous ions.** The structure of the product ion is another key issue for mass spectrometry. The radius of gyration ($R_{gyr}$) represents the compactness. The evolution of the $R_{gyr}$ of the nanodisc shows a sharp increase in 5/10 simulations, but a slight decrease in the remaining 5 simulations (Fig. 3b). The sharp increase of $R_{gyr}$ indicates dramatic conformational reorganization in a short simulation time (tens of nanoseconds) (Figs. 1–2 and S3–S10) occurs where and when the solvent shell is initially lost.

The calculated collision cross sections (CCSs) of the product gaseous ions are generally consistent with their $R_{gyr}$ (Fig. S12). Compact molecules present lower CCS than elongated ones. The calculated CCSs of ten product ions range from 80 to 140 nm$^2$ (Fig. 3c and Table 1), while the initial structure has a CCS of 104.1 nm$^2$. The deviation of the CCSs of the product ions from that of the initial structural model is consistent with what observed for nanodiscs and some other proteins experimentally[46,47]. For example, the calculated CCS for a MSP1D1-DMPC nanodisc model is 85.6 nm$^2$ with the

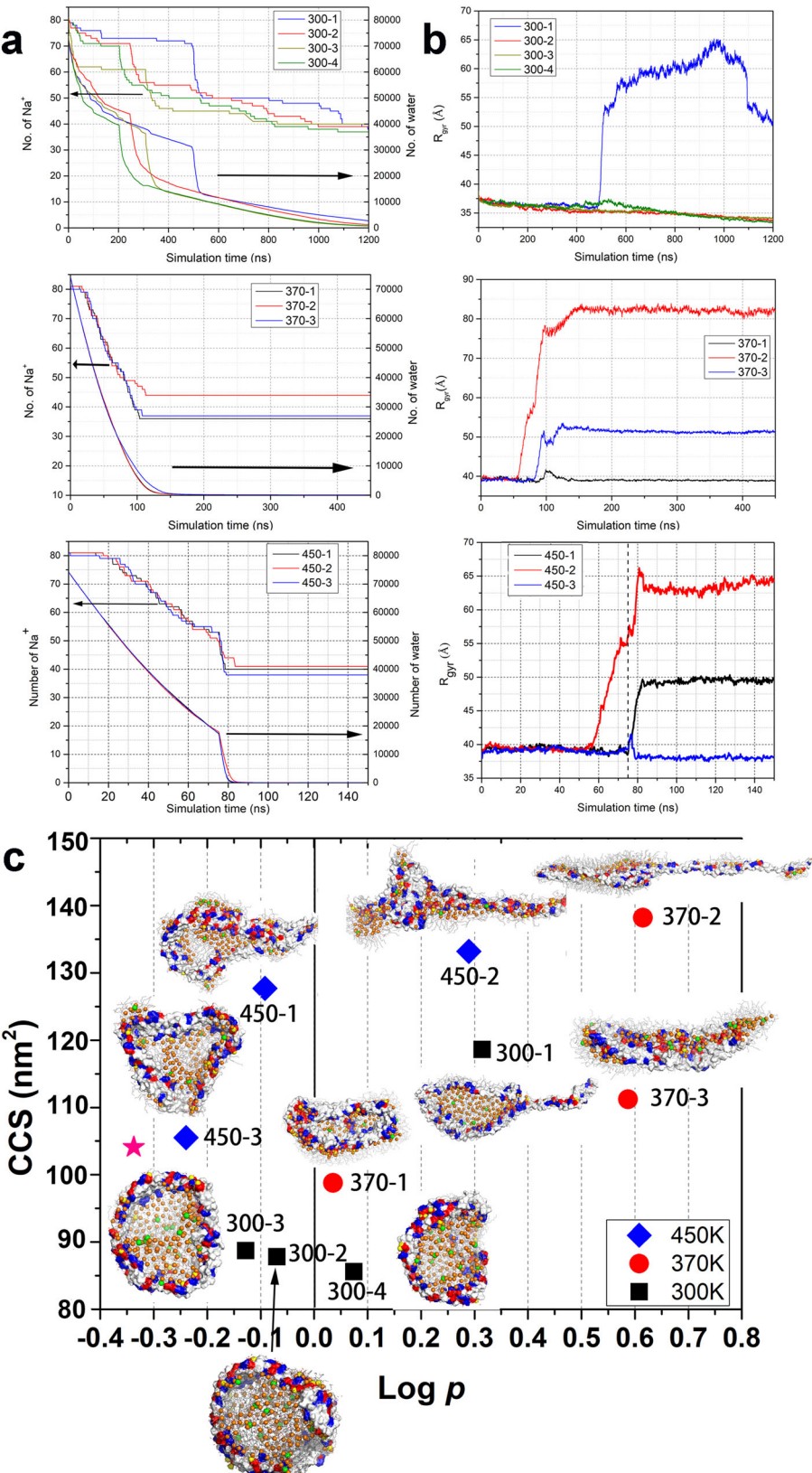

**Fig. 3 Evolution of charged nanodisc droplet in all ten simulations and the diversity of the product gaseous ions. a, b** The evolutions of the number of water and Na$^+$ molecules in remaining nano-droplets, and radius of gyration ($R_{gyr}$) of the nanodisc in all simulations at 300, 370, and 370 → 450 K from top to bottom. Black arrows in panel A indicate the corresponding axis of the lines. **c** The morphologies of the product gaseous ions, which were projected on the collision cross section (CCS) and log ($p = \frac{a/b}{b/c}$), where a, b and c are the lengths of three principal axes of inertia tensors. The MSPs are displayed in surface representation, with positively and negatively charged residues colored in blue and red, respectively. The phosphorus atoms are shown in orange spheres, while Na$^+$ ions in yellow (binding on protein surface) and green spheres. The asterisk in hot pink marks the initial structure.

**Table 1 Properties of all product gaseous ions.**

| Simulation | Compositions of product ions | m/z | Shape | | | Charge states | |
|---|---|---|---|---|---|---|---|
| | | | CCS (nm$^2$) | a/b | b/c | λ | β-λ |
| 300-1[b] | $[2MSP1 + 183DMPC + 38Na + 2701H_2O]^{26+}$ | 8467 | 118.6 | 2.67 | 1.29 | 29 | 9 |
| | $3[1DMPC]^0$ | - | - | - | - | - | - |
| 300-2[a] | $[2MSP1 + 186DMPC + 39Na + 1262H_2O]^{27+}$ | 7269 | 87.8 | 1.13 | 1.33 | 22 | 17 |
| 300-3[a] | $[2MSP1 + 186DMPC + 40Na + 710H_2O]^{28+}$ | 6655 | 88.7 | 1.07 | 1.44 | 26 | 14 |
| 300-4[a] | $[2MSP1 + 186DMPC + 37Na + 839H_2O]^{25+}$ | 7544 | 85.6 | 1.34 | 1.13 | 22 | 15 |
| 370-1[a] | $[2MSP1 + 186DMPC + 26Na + 99H_2O]^{24+}$ | 7302 | 98.8 | 1.81 | 1.67 | 31 | 5 |
| 370-2[b] | $[2MSP1 + 186DMPC + 44Na + 75H_2O]^{32+}$ | 5469 | 138.2 | 6.27 | 1.52 | 39 | 5 |
| 370-3[a] | $[2MSP1 + 186DMPC + 37Na + 94H_2O]^{25+}$ | 7007 | 111.2 | 3.98 | 1.03 | 24 | 13 |
| 450-1[a] | $[2MSP1 + 186DMPC + 40Na + 1H_2O]^{28+}$ | 6200 | 127.7 | 2.01 | 2.48 | 31 | 9 |
| 450-2[b] | $[12DMPC + 3Na + 4H_2O]^{3+}$ | 2759 | - | - | - | - | - |
| | $[2MSP1 + 174DMPC + 41Na + 6H_2O]^{29+}$ | 5708 | 133.2 | 3.28 | 1.68 | 35 | 6 |
| 450-3[a] | $[1DMPC + 1Na + 8H_2O]^{1+}$ | 845 | - | - | - | - | - |
| | $[5DMPC + 2Na + 38H_2O]^{2+}$ | 2060 | - | - | - | - | - |
| | $[2MSP1 + 180DMPC + 38Na]^{26+}$ | 6517 | 105.5 | 1.10 | 1.92 | 31 | 7 |

[a]The simulation follows the at-center process; [b]the simulation follows the off-center process.

experimental value of 89.2 ± 8.0 nm$^2$, which corresponds to the product gaseous ions[47].

To describe the shapes of the product ions, we calculated their inertia tensors to get the length of the three principal axes a, b, and c (from longest to shortest, Fig S13). The logarithm of the ratio, log ($p = \frac{a/b}{b/c}$), was defined to characterize the shape of the product ions. Log $p = 0$ indicates a preferred sphere, log $p < 0$ an oblate shape, and log $p > 0$ a prolate shape. The value of log $p$ of our product ions is distributed between −0.30 to 0.70 with log $p = -0.34$ (a/b = 1.03, and b/c = 2.20) for the initial conformation, which appears a distinctly oblate shape. Projection of all product gaseous ions on CCS and log $p$ clearly indicates the diversity of the product shapes. We thus argue that the observed lack of correlation between CCS and log $p$ suggests that CCSs do not distinguish different structures very well, especially for macromolecular complexes.

Compared to the shape of the initial structure, the product ions with lower CCS collapse into a more spherical shape, such as the product ions of the 300-2, 300-3, 300-4, 370-1, and 450-3 simulations, all of which follow the at-center process. The exceptional case, the product ion of the 450-1, has a small log $p$ of −0.09, but a large CCS of 127.7 nm$^2$, since the values of a/b and b/c are relatively large (Table 1). The remaining simulation of the at-center group, the 370-3 simulation, yields a distinctly prolate shape with a log $p$ of 0.59, but a CCS of 111.2 nm$^2$. The feature of the off-center process, the extrusion of one MSP monomer, produces large log $p$ and CCS (Fig. 3c).

**Structural rearrangements of product gaseous ions**. The lipid bilayers collapse to varying degrees after the ESI process. From the structures of the product gaseous ions (Figs. 3c and S14–16), we found two different arrangements of DMPC lipids that lower the energy of product ions in the gas phase. Some lipid molecules maintain a bilayer, while some form monolayers covering the hydrophilic surface with their tails exposed to the gas phase, to minimize the surface tension. In the latter, lipids prefer to be clustered together, covering the polar surface of protein, solvent molecules, and lipid bilayer.

The ESI process also results in dissociation of the MSP dimer to different extent (Figs. 3c and S14–16). After the at-center processes, both monomers of the MSP dimer remain circular, but the original planar structure folds into solid, more compact, and spherical configurations. Conversely, one monomer of the MSP dimer in the off-center processes stretches from a ring into a straight chain.

**The mass-to-charge (m/z) ratios**. The m/z ratios of all product ions spread in a wide range of 5500–8500 (Table 1). The wide distribution of m/z is consistent with the mass spectrum of nanodiscs[48]. The three offspring lipid-Na$^+$ ions at high temperature vary in their compositions, charge states and m/z ratios (Table 1). The range of their m/z ratios also matches the peaks at low m/z[48].

The gaseous ions are almost devoid of water, comprised of two MSPs, α lipids and β Na$^+$ ions: $[2MSP1 + \alpha DMPC + \beta Na]^{(\beta-12)+}$. At low temperatures, however, a considerable amount of water molecules remains bound due to the limited simulation time (Table 1). In some simulations, the MSPs may be partially extruded, but none detach completely. The number of remaining lipids is a variable, and may be the factor that has the greatest impact on the total mass. However, in all simulations, the loss of lipids does not exceed 12 (Table 1). That is to say, the $m$ has a limited effect on the m/z ratio, and then $z$ should have a greater impact.

The ESI charge states of the product ions are widely distributed, ranging from 24 to 32. The range is reasonable compared with the range inferred from results of the mass spectrometry of MSP1D1 nanodiscs (20–25)[41,42], which is 11 residues less than the MSP1 used in our simulations. What determines the ESI charge states ($z$), i.e., the number (β) of residual Na$^+$ ions in the product ions ($z = \beta-12$)? From the structures of the gaseous ions (Figs. 3c and S14–16), we found that some of the sodium ions (λ) appear tightly bound on the protein surface. These sodium ions are hard to release, while the rest (β-λ) form lipid-Na$^+$ clusters which could be released into vacuum, as seen in simulations 450-2 and 450-3. Therefore, the charge state can be considered to be composed of two components: $[2MSP1 + \lambda Na]^{(\lambda-12)+} + [\alpha DMPC + (\beta-\lambda)Na]^{(\beta-\lambda)+}$. The first term could be considered as the accommodation of Na$^+$ on the MSP surface, depending on the conformation of the MSP dimer. The extended MSP conformations from the off-center processes hold more sodium ions, and present stronger protein-Na$^+$ electrostatic energies than those from other simulations at the same temperatures (Table 1 and Figs. S17–S19). The remaining number of Na$^+$ ions, the second term, interacting with lipids only, by contrast, is more variable, because they can be released from the droplet in the ESI process. Therefore, the interplay of the various conformations

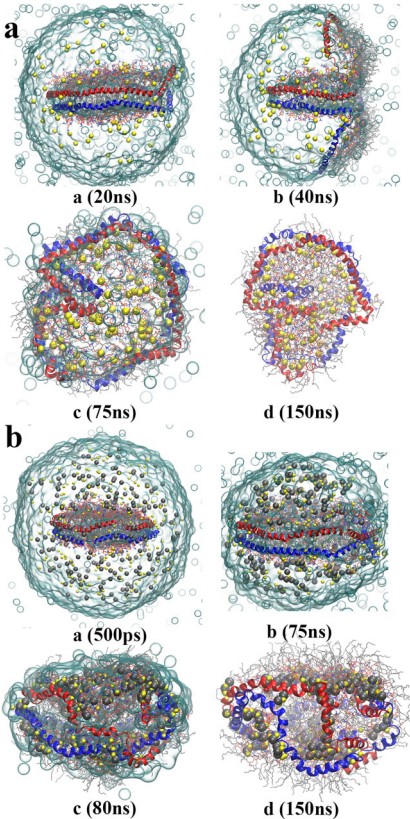

**Fig. 4 The importance of long-range electrostatic interactions. a, b** Evolution of the charged nanodisc droplet in the C12 and neutral simulations respectively. The representation is the same as in Fig. 1. The compositions of the product gaseous ions are $[2MSP1 + 186DMPC + 81Na + 3H_2O]^{69+}$ and $[2MSP1 + 186DMPC + 196Na + 183Cl + 4H_2O]^{1+}$ respectively.

of MSP and the diversity of the arrangement and number of lipids definitely makes product ions and mass-to-charge ratios diverse, which may be the explanation for the observed experimental spread.

**The importance of long-range electrostatic interactions**. The morphologies of product ions and details of the ESI process strongly depend on electrostatic interactions. To test the importance of the treatment of long-range electrostatic interactions, we carried out two simulations using a cutoff to calculate the electrostatic interactions (cutoff = 12 and 33 Å, termed C12 and C33, respectively), without long-range corrections (Table S1). Both simulations follow the off-center scenario, but without partial extrusion of MSP monomers. The product gaseous ions form a reverse micelle, with hydrophobic lipid tails toward the vacuum, wrapping the MSPs and sodium ions (Figs. 4a and S20–S21). No release of lipid oligomers was observed in both simulations, and the reduction of ejections of $Na^+$ resulted in the high charge states of the product gaseous ions (69+ and 42+, respectively). Compared with the previous simulations, it was demonstrated that the strong electrostatic interaction results in the partial extrusion of MSP monomers.

To confirm the key role of electrostatic repulsions among net charges, one more simulation (termed neutral) was performed, in which the nanodisc was solvated with a 150 M NaCl aqueous solvent (213 $Na^+$ and 201 $Cl^-$) without net charge (Table S1). This neutral simulation follows the at-center process (Figs. 4b and S21), yielding a spherical gaseous reverse micelle, and most of the

ions remain in the gaseous product. The nanodisc stays more stably at the droplet center than in simulations with net charge, indicating that the deviation of the nanodisc from the center of the nano-droplet is induced by the electrostatic repulsion.

## Discussion

This work provides the first atomistic view of lipid nanodiscs releasing from highly charged nano-droplets into the gas phase. Our simulations suggest that gaseous nanodisc ions are formed by evaporating the solvent molecules to dryness, with two distinct scenarios. The at-center and off-center process comply with the previously proposed CRM model and a CRM/CEM hybrid model, respectively. The CRM processes produce compact gaseous ions, while the CRM/CEM processes yield extended conformations with partially extruded MSP monomers. The formation of gaseous ions via various competing mechanisms has also been reported previously in the ESI process of polymers[19,27,49], and compact proteins unfolding during release[7].

According to Rayleigh's macroscopic model, the energy of the charged droplet is the sum of electrostatic repulsion and surface energy[50]. The surface tension prefers a spherical shape, while the electrostatic repulsion tends to destabilize the spherical shape. In the at-center scenario, the droplet maintains a nearly spherical shape, especially when there are enough water molecules to keep the solvent shell of the elliptical nanodisc, indicating that this scenario is driven by the surface tension. It was further confirmed by the simulations without electrostatic repulsions, which produced a stable at-center process. In contrast, in the off-center process, one monomer of MSP is extruded partially from the nano-droplets, and the off-center processes show slower $Na^+$ ejection than the at-center cases do, especially obvious in the comparison of the simulations at 300 K (Fig. 3a). In addition, the simulations with weakened electrostatic repulsion follow the off-center process without partial extrusion of MSP monomers. It demonstrates that the off-center trajectories are driven by electrostatic repulsion.

Ultimately, nano-droplets morph into almost dried-out nanodisc complexes after solvent evaporation and Coulomb fission of small ions. Some solvated lipid-$Na^+$ ions were released in this process, which correspond to the peaks at low $m/z$[41]. No detachment of the MSP monomer from the nanodisc complex was observed in any simulation, which is also consistent with experimental observations. In all simulations, the bilayer is collapsed to some degree, with varying levels of dissociation of the MSP dimer, which is elevated in simulations at higher temperatures. This collapse may benefit the release of an embedded membrane protein if present. Mass spectrometry experiments also show that high collisional activation is required to liberate the embedded membrane protein[32,51,52].

According to the distribution of charge carriers $Na^+$ in the product ions, the residual sodium ions are bound chiefly on the MSP surface, while the rest interact with lipid clusters. The ESI charge states are governed by the number of remaining lipids and the conformation of the MSPs. Both of them are complicated and diverse as shown in our simulations. The interplay of these two factors may results in the reported broad mass spectra of nanodiscs. The calculated $m/z$ data agree with the range of the mass spectra. However, due to limits on computational resources and the expensive cost of ESI simulations, it is currently infeasible to perform enough replicas of simulations to statistically compare the $m/z$ ratios from MD simulations with the experimental mass spectra.

The mass of the product ion ($m$) is mostly determined by the loss of lipids in the ESI process. Ejection of solvated sodium ions has to overcome the surface tension, but ejection of solvated

lipid-sodium complex ions has to overcome the additional hydrophobic interactions of lipid tails and polar interactions of lipid heads caused by lipid packing. The released lipid-sodium complex ions in our simulations are loosely packed in the droplets. Meanwhile, we have also observed evaporation of lipids without sodium ions due to weak interactions at the tail of the expelled MSP monomer. Furthermore, some tightly bound water molecules and those wrapped by lipids may be prevented from evaporating. Conversely, all these factors make it difficult to predict the mass of the product ion.

The ESI charge states ($z$) from MD simulations may be affected by the approach to calculate the electrostatic energy and the accuracy of the force field. Previous studies predicted the ESI charge states of globular proteins quite accurately[28]. In the case of globular proteins, the only pertinent variable is the effective protein radius[10], which is relatively constant during the ESI process. We employed the PME method[53] to calculate the long-range electrostatic interactions here, while previous simulations[22,28] were carried out without cutoffs, which is infeasible for the large nanodisc droplet with a radius larger than 8 nm. The artifacts of PME have been studied in previous studies. For a non-neutral system, the use of PME implicitly introduces a homogeneous background charge to neutralize the system[54,55]. The background charge may weaken the electrostatic interaction to some extent, which is strongly dependent on the unit cell size[54,55]. Therefore, in our cases, with such a large unit cell size (1000 Å × 1000 Å × 1000 Å), the effect should be very slight. To test the accuracy of PME, we repeated the reported ESI simulations of Ubq[28] with PME and obtained a very similar ESI process and the same ESI charge states (Fig. S22). Thus, we believe that the atomistic ESI process revealed in this study would not be unduly influenced by different approaches of calculating electrostatic energies. One practical difficulty is the limitation of the computational resource to use polarizable force field for large systems and long time-scale simulations, as the force field used in this study with fixed partial charges may not reflect the charge redistribution caused by the net charge. Another limitation may arise from using of the force field, parametrized for the room temperatures and in the liquid phase, in the MD simulations at high temperatures (370 and 450 K), and in the gas phase. However, on one hand, the release of nanodisc from the nano-droplet obtained from the MD simulation at high temperature is consistent with that at room temperature (300 K). On the other hand, in reported studies, MD simulations at high temperature produced the same $m/z$ ratios as the mass spectrometry experiments[21,28,29], and the force field for the liquid phase is suitable for modeling gaseous ions[56]. Therefore, the limitations of the methods used in this work may affect the accuracy of the predicted $m/z$ ratio, but likely have little influence on the observed ESI process.

The structural model of MSP1-DMPC nanodisc we constructed in this study, shares the same discoid shape with all nanodiscs, which just vary in size and the lipid composition, and yield very similar patterns of mass spectra[43,48]. It indicates the consistency in molecular mechanism of the ESI process for all nanodiscs. When membrane proteins are embedded in the nanodiscs, membrane proteins tend to disrupt the stability and lateral packing of the nanodiscs[57,58], and may enhance the collapse propensity at low simulation temperature. Work for systems with membrane proteins in the nanodisc is ongoing. Therefore, we believe that the release of nanodisc from the charged nano-droplet, obtained from our MD simulations, could be generalized to all nanodiscs, protein-loaded and protein-free.

To conclude, our simulations provide a clear picture of the behavior of nanodiscs in the ESI process, the conformational rearrangement from liquid to gas phase, and various structures of product gaseous ions. This work may provide structural support for the interpretation of mass spectra of complicated systems, especially protein-lipid systems.

## Methods

**System setup**. The initial structure was built in CharmmGUI[59], containing two MSP1 proteins and 186 DMPC lipids (Fig. S1). All the titratable residues were in their default protonation states at pH 7.0. The upper limit of net droplet charge is according to the Rayleigh limit[50]

$$Z_R = \frac{8\pi}{e} \sqrt{\varepsilon_0 \gamma R^3}$$

where $\varepsilon_0$ is the vacuum permittivity, $\gamma$ is the surface tension (0.05891 Nm$^{-1}$ for water at 370 K), $R$ is the droplet radius (~8 nm for this system), so $Z_R \approx 81$. 85% of the Rayleigh limit (69) was used for the net charge of the droplet. The ESI usually generates protonated ions, but changing protonation states are impracticable due to fixed charges in standard MD protocols. The protonated and sodiated states show similarities in charge states and collision cross sections[28]. Thus, sodium ions were used as the charge carriers as protons are difficult to treat in MD simulations.

**Simulation details**. The CHARMM36 force field[60] was employed for all simulations. Different force fields, CHARMM36 and OPLS-AA, showed only slight difference in ESI charge states[28]. TIP3P water modified for the CHARMM force field was used[61]. Different water models result in very similar evaporation processes[62] and very similar behavior of the analytes[28]. All simulations were carried out in NAMD2.10[63]. The minimized systems from CharmmGUI were equilibrated in six steps, gradually reducing restraints applied on the positions of proteins and lipids and the dihedral angles of proteins, followed by 1 ns runs without any restraints. For the equilibration step, the Langevin thermostat and barostat[64] with a relaxation time of 1.0 ps were employed to maintain a temperature of 300 K and a pressure of 1.0 bar. The nonbonded interactions were truncated at 12 Å with a shift at 10 Å. The Particle Mesh Ewald (PME)[53] method was used to calculate the long-range electrostatic interactions. The hydrogen-heavy atom bonds were restrained by the SHAKE algorithm[65] to allow a time step of 2 fs.

The following simulations of the ESI process were performed in the NVT ensemble. The temperature was kept at 300 K, 370 K, and 370 → 450 K (370 K for the initial 75 ns, and 450 K for the last 75 ns) by Langevin thermostat (Table S1). The equilibrated systems were located at the center of a cubic vacuum box of 1000 Å × 1000 Å × 1000 Å (Fig. S1), which was large enough to avoid interactions between adjacent images in the evaporation. The long-range electrostatic interactions were calculated using the PME method. These simulations are very resource-consuming (Table S2). To speed up the simulations, the box size was reduced to 500 Å × 500 Å × 500 Å when less than two water molecules were evaporated in a 250 ps run. A detailed description of the treatment of nonbonded interactions is available in the supplementary methods.

The trajectory stitching approach[15,22] was employed. Each production simulation was divided into a set of 250 ps short runs. After each run, the evaporated species that moved more than 70 Å away from the nanodisc surface were removed. Then new velocities were assigned for the next 250 ps simulation.

**Simulation analysis**. VMD[66] was used for visualization and data analysis. The lengths along the three principal axes of inertia of moments were noted as $a$, $b$, and $c$ from longest to shortest. The ratio p was calculated by $= \frac{a}{b} \div \frac{b}{c}$, where $p < 1$ indicates an oblate shape, $p = 1$ means a sphere, and $p > 1$ represents a prolate shape. The collision cross section (CCS) was calculated by the trajectory method using the Mobcal program[67,68]. We calculated the inertia of moments and CCSs for the last frames of all ten simulations, and the last frame means the conformation at the simulation time of 1200, 450, and 150 in the simulations at different temperatures: 300, 370, and 370 → 450 K respectively (Table S1).

## Data availability

Data that support this study are available from the corresponding authors upon reasonable request. The structural coordinates of the initial and final conformations of our simulations are available at https://github.com/shabeir/ESI-of-nanodics.

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

## Acknowledgements

This work was supported by the National Natural Science Foundation of China (No. 31971176 and 31800616), the Natural Sciences and Engineering Research Council (Canada), the Canada Research Chairs program. Calculations were carried out on Compute Canada resources.

## Author contributions

B.W. and D.P.T. designed the simulations and wrote the paper. B.W. performed and analyzed the simulations.

## Competing interests

The authors declare no competing interests.
