## [Peer Review File · Communications Chemistry]

Reviewers' comments:

Reviewer #1 (Remarks to the Author):

This manuscript reports some excellent molecular dynamics simulation data on the behavior of lipid nanodiscs in highly charged electrospray droplets. The system size explored here is unprecedented and most impressive, and the simulation results match many of the experimentally observed features (such as charge states and collision cross sections). Most importantly, these data for the first time provide atomistic insights into the behavior of nanodiscs under electrospray conditions. Although the science is great, the writing is a bit bumpy in some sections. However, with a bit of editorial polishing this will be easy to fix. I ask the authors to consider the following suggestions when preparing the final version of their manuscript.

Overall, I enthusiastically support publication of this work in COMMSCHEM.

Specific Comments:

The paper is quite short, and extensively refers to the (very large) SI file. Most readers will never look at the SI. It would be nice to transfer some of the SI figures to the main text.

It appears that these ESI simulations use by far the largest droplets to date. Larger droplets may have been simulated by some, but not on such very long time scales. This impressive fact should be clearly stated.

p. 3: "The evaporated molecules are removed by collisional activation" not clear. Presumably, the authors want to say that residual solvent molecules adhering to the gaseous analyte ions are removed by collisional activation.

p. 3: "It has been proposed, but with controversies, that the three models are applicable for different molecules: IEM for small ions, ..." To highlight the differences in opinion related to this question, the authors should also cite ref 47 after this sentence (demonstrating IEM for relatively large proteins).

p. 3: "monodisperse and homogeneous" this seems redundant

p. 3: "suggested to arise from isolated" change to "arising from isolated" (the original wording indicates that the nature of these signals is not clear). Similarly, on p. 8: "which could correspond to the peaks at low m/z40." The chemical nature of these ions will be unambiguous from their mass, there is no need to speculate.

p.3/p.4. When describing earlier MS experiments on nanodiscs, the key point is to emphasize more clearly that the discs undergo partial disassembly during ESI. This will set the stage for the MD results. (the present wording with "sharp peaks" etc. is a bit nebulous)

p. 4: "may not be as simple as that for homogeneous systems" change to "may not be as simple as that for experiments in bulk solution"?

When starting the Results section, the authors have to introduce their system. This section should be comprehensible without consulting the Methods chapter. Droplet radius, system charge, intrinsic protein charge, temperatures (e.g., the cryptic 300-2 etc. notation is never really explained, although readers can guess), number of water molecules, etc. ...

Why were different temperatures used, and what were the differences seen for these different temperatures? Presumably, water evaporation is much faster at higher temperature? Some of this information is buried in the SI, but it is not very accessible.

p. 5/6: "We projected ... for macromolecular complexes." This section is poorly written. Please improve.

Perhaps I missed it, but it seems that the method/software used for CCS calculations is not specified.

p. 5: "between theoretical and experimental CCS 43, 44." Please be more specific: how do the experimental and MD data compare? Provide numbers.

p. 6: "The range is rational compared with the speculated range in the mass spectrometry" please improve language. "rational" and "speculated" have to be replaced.

p. 7: "The morphologies of product ions and details of the ESI process strongly depend on electrostatic interactions...." This entire paragraph should be moved to RESULTS. The lessons learned from the charged vs neutral runs should be explained more clearly.

p. 8: "The ESI charge states (z) from MD simulations may be affected ..." This sections has to be moved from DISCUSSION to RESULTS as well.

p. 8: "if presents" typo

p. 8: "Mass spectrometry experiments also show ..." the general concept of using nanodiscs for generating relatively clean mass spectra of membrane proteins should have been more clearly explained in the Introduction.

Reviewer #2 (Remarks to the Author):

Editorial note: this Reviewer provided comments to the Editor only.

Reviewer #3 (Remarks to the Author):

In this paper, the mechanics of electrospray ionization process of a lipid nanodisc is studied via atomistic molecular dynamics simulation. The authors claim to provide a detailed atomistic view of the process and, therefore, provide helpful insights for the MS community to better interpret mass spectra of lipid nanodiscs or other protein-lipid complexes. They have observed variability in shapes obtained by processes that agree with either the CRM (charge-residue model) or the CEM (chain ejection model) mainly.

Even though there is already some published work on protein ESI, the study of nanodisc ESI process via molecular dynamics simulation is new according to my knowledge. It would have been also interesting to see what happens when a membrane protein is embedded in the nanodisc, does it stay in the nanodisc or would it be expelled from it during the ESI process?

Given that this is a purely computational work, special care should be taken to choose an appropriate model. The authors should clarify the reason to run simulations at high temperatures such as 450K given that neither the water nor the protein model are parametrized for those temperatures. Is there a difference between experimental water surface tension values at the simulated temperatures and CHARMM modified-TIP3P that could affect the dynamics of solvent evaporation?

In general terms, the methods are described appropriately so that it this work can be reproduced. However, when reporting a computed value, more detailed information is missing. E.g in Figure S13 (CCS vs Rgyr) there are no errors assigned to any of the values. How were they computed? Coming from 1 frame, an average taken over the whole production run, over a certain number of last frames? This should be clarified.

Could the authors mention how generalizable are these results to other nanodisc compositions?

Minor clarifications:

In the methods section, the following phrase is written: "To speed up the simulations, the box size was reduced to $500\text{\AA} \times 500\text{\AA} \times 500\text{\AA}$ when less than two water molecules were evaporated in each run." Each run here means after the 150ns equilibration?

Following that, In the Table S2 (Simulation performance speeds for different methods, why do the systems have different number of atoms? E.g Cutoff+PME, box 100nm= 150105 vs Cutoff+PME box 50nm = 27972 if they come from the equilibrated 100nm size box?

In Figure S21, maybe the lines corresponding to number of waters and number of Na+ could have different styles to help the reader, for instance continuous and dashed.

In summary, this paper gives a structural overview about the ESI process on MSP1-DMPC nanodiscs, and can be of interest for the MS and biophysics community

Response to Reviewers' comments:

We would like to thank the reviewers for their careful reading and thoughtful comments to improve the manuscript. We appreciate the overall positive tone and made every effort to address the issues raised in the reviews as detailed below:

Reviewer #1 (Remarks to the Author):

This manuscript reports some excellent molecular dynamics simulation data on the behavior of lipid nanodiscs in highly charged electrospray droplets. The system size explored here is unprecedented and most impressive, and the simulation results match many of the experimentally observed features (such as charge states and collision cross sections). Most importantly, these data for the first time provide atomistic insights into the behavior of nanodiscs under electrospray conditions. Although the science is great, the writing is a bit bumpy in some sections. However, with a bit of editorial polishing this will be easy to fix. I ask the authors to consider the following suggestions when preparing the final version of their manuscript.

Overall, I enthusiastically support publication of this work in COMMSCHEM.

Specific Comments:

The paper is quite short, and extensively refers to the (very large) SI file. Most readers will never look at the SI. It would be nice to transfer some of the SI figures to the main text.

Response: We appreciate the thoughtful comment. Some figures were transferred from the SI to Figures 3 and 4 in the main text.

It appears that these ESI simulations use by far the largest droplets to date. Larger droplets may have been simulated by some, but not on such very long time scales. This impressive fact should be clearly stated.

Response: We appreciate the thoughtful comment. It was stated on Page 4.

p. 3: "The evaporated molecules are removed by collisional activation" not clear. Presumably, the authors want to say that residual solvent molecules adhering to the gaseous analyte ions are removed by collisional activation.

Response: We apologize for this mistake. What we're trying to say here is "the evaporated neutral solvents are carried away by the heated carrier gas". It was modified on Page 3.

p. 3: "It has been proposed, but with controversies, that the three models are applicable for different molecules: IEM for small ions, ..." To highlight the differences in opinion related to this question, the authors should also cite ref 47 after this sentence (demonstrating IEM for relatively large proteins).

Response: We appreciate the reminder. The reference was added on Page 3.

p. 3: "monodisperse and homogeneous" this seems redundant

Response: We agree with the reviewer. "and homogeneous" was deleted.

p. 3: “suggested to arise from isolated” change to “arising from isolated” (the original wording indicates that the nature of these signals is not clear). Similarly, on p. 8: “which could correspond to the peaks at low m/z 40.” The chemical nature of these ions will be unambiguous from their mass, there is no need to speculate.

Response: We appreciate the thoughtful comment. The sentences were rephrased on Page 3-4 and Page 9.

p.3/p.4. When describing earlier MS experiments on nanodiscs, the key point is to emphasize more clearly that the discs undergo partial disassembly during ESI. This will set the stage for the MD results. (the present wording with “sharp peaks” etc. is a bit nebulous)

Response: The sentences were rephrased on Page 4.

p. 4: “may not be as simple as that for homogeneous systems” change to “may not be as simple as that for experiments in bulk solution”?

Response: The sentences were rephrased on Page 4.

When starting the Results section, the authors have to introduce their system. This section should be comprehensible without consulting the Methods chapter. Droplet radius, system charge, intrinsic protein charge, temperatures (e.g., the cryptic 300-2 etc. notation is never really explained, although readers can guess), number of water molecules, etc. ...

Response: The beginning of the Results was improved with more details on Page 4.

Why were different temperatures used, and what were the differences seen for these different temperatures? Presumably, water evaporation is much faster at higher temperature? Some of this information is buried in the SI, but it is not very accessible.

Response: Different temperatures were used, because the temperature of the late stage of the ESI process is still controversial. Some studies have demonstrated it is at room temperature, while some studies supported higher temperatures. The water evaporation is much faster at higher temperature, but the behaviors of nanodiscs at different temperatures are consistent. This information was added on Page 4.

p. 5/6: “We projected ... for macromolecular complexes.” This section is poorly written. Please improve.

Response: This section was rewritten on Page 6.

Perhaps I missed it, but it seems that the method/software used for CCS calculations is not specified.

Response: The details of the method/software for the CCS calculation was added in the Methods on Page 11.

p. 5: “between theoretical and experimental CCS 43, 44.” Please be more specific: how do the experimental and MD data compare? Provide numbers.

Response: Specific information was added on Page 6.

p. 6: “The range is rational compared with the speculated range in the mass spectrometry” please improve language. “rational” and “speculated” have to be replaced.

Response: The sentences were rephrased on Page 7.

p. 7: “The morphologies of product ions and details of the ESI process strongly depend on electrostatic interactions....” This entire paragraph should be moved to RESULTS. The lessons learned from the charged vs neutral runs should be explained more clearly.

Response: This section was moved to Results, and explained more clearly on Pages 7-8.

p. 8: “The ESI charge states (z) from MD simulations may be affected ...” This sections has to be moved from DISCUSSION to RESULTS as well.

Response: We think this section is a discussion of the limitation of the method, so we would like to keep it in the Discussion.

p. 8: “if presents” typo

Response: It was corrected on Page 9.

p. 8: “Mass spectrometry experiments also show ...” the general concept of using nanodiscs for generating relatively clean mass spectra of membrane proteins should have been more clearly explained in the Introduction.

Response: It was explained on Pages 3-4.

Reviewer #2 (Remarks to the Author):

No comments.

Reviewer #3 (Remarks to the Author):

In this paper, the mechanics of electrospray ionization process of a lipid nanodisc is studied via atomistic molecular dynamics simulation. The authors claim to provide a detailed atomistic view of the process and, therefore, provide helpful insights for the MS community to better interpret mass spectra of lipid nanodiscs or other protein-lipid complexes. They have observed variability in shapes obtained by processes that agree with either the CRM (charge-residue model) or the CEM (chain ejection model) mainly.

Even though there is already some published work on protein ESI, the study of nanodisc ESI process via molecular dynamics simulation is new according to my knowledge. It would have been also interesting to see what happens when a membrane protein is embedded in the nanodisc, does it stay in the nanodisc or would it be expelled from it during the ESI process?

Response: In this study, we just reported the work for the lipid nanodiscs, while the work for a membrane protein embedded in the nanodisc is ongoing, and will be reported later.

Given that this is a purely computational work, special care should be taken to choose an appropriate model. The authors should clarify the reason to run simulations at high temperatures

such as 450K given that neither the water nor the protein model are parametrized for those temperatures. Is there a difference between experimental water surface tension values at the simulated temperatures and CHARMM modified-TIP3P that could affect the dynamics of solvent evaporation?

Response: We agree with the reviewer. The force field used in our simulation is not parametrized for high temperatures. The protocol of simulations at high temperatures (450K) is following the previous studies, in which the simulations obtained the same m/z as the experiments. And the release of nanodisc from the nano-droplet obtained from the MD simulation at high temperature is consistent with that at room temperature (300K). Therefore, we think that the force field is suitable for this MD simulation of the ESI process.

The water model of CHARMM modified-TIP3P results in lower surface tension than the experimental values at all temperatures. However, previous work has demonstrated that different water models, result in different surface tensions, exhibit very similar evaporation process. Moreover, the CHARMM force fields for proteins and lipids are fine-tuned to be compatible with the water model of CHARMM modified-TIP3P. Therefore, we think that the CHARMM modified-TIP3P is adequate for our work. In future work we will also consider TIP4P/2005, which has a better surface tension but is less tested with proteins and lipid parameters.

The explanations were added in the Methods (Page 11) and Discussion (Page 10).

In general terms, the methods are described appropriately so that it this work can be reproduced. However, when reporting a computed value, more detailed information is missing. E.g in Figure S13 (CCS vs R_{gyr}) there are no errors assigned to any of the values. How were they computed? Coming from 1 frame, an average taken over the whole production run, over a certain number of last frames? This should be clarified.

Response: The radius of gyration (R_{gyr}) and the collision cross sections (CCSs) in Figure S13 were calculated from the last frames of the ten simulations. And the details were added in the Methods on Page 11 and the legend of Figure S13.

Could the authors mention how generalizable are these results to other nanodisc compositions?

Response: The structural model of MSP1-DMPC nanodisc, we constructed in this study, shares the same discoid shape with all nanodiscs, which just vary in size and the lipid composition, and yield very similar pattern of mass spectra. It indicates the consistency in molecular mechanism of the ESI process for all nanodiscs. When membrane proteins are embedded in the nanodiscs, membrane proteins tend to disrupt the stability and lateral packing of the nanodiscs, and may enhance the collapse propensity at low simulation temperature. The work for the system with a membrane protein embedded in the nanodisc is ongoing. Therefore, we believe that the release of nanodisc from the charged nano-droplet, obtained from our MD simulations, could be generalized to all nanodiscs, protein-loaded and protein-free.

It was added in the Discussion on Page 10.

Minor clarifications:

In the methods section, the following phrase is written: "To speed up the simulations, the box size was reduced to $500\text{\AA} \times 500\text{\AA} \times 500\text{\AA}$ when less than two water molecules were evaporated in each run." Each run here means after the 150ns equilibration?

Response: We apologize for this confusing statement. Each run here means after a short run of 250 ps. It was corrected in the Methods on Page 11.

Following that, In the Table S2 (Simulation performance speeds for different methods, why do the systems have different number of atoms? E.g Cutoff+PME, box 100nm= 150105 vs Cutoff+PME box 50nm = 27972 if they come from the equilibrated 100nm size box?

Response: The trajectory stitching approach was employed in this study. Each production simulation was divided into a set of 250 ps short runs. After each run, the evaporated species that moved more than 70 Å away from the nanodisc surface were removed. The different number of atoms results from the evaporation of solvent molecules. It was added in the Methods on Page 11.

In Figure S21, maybe the lines corresponding to number of waters and number of Na⁺ could have different styles to help the reader, for instance continuous and dashed.

Response: Arrows were added to indicate the corresponding axis of different lines in Figure S21.

In summary, this paper gives a structural overview about the ESI process on MSP1-DMPC nanodiscs, and can be of interest for the MS and biophysics community.

REVIEWERS' COMMENTS:

Reviewer #1 (Remarks to the Author):

The authors have addressed all of the reviewers' comments. The paper can be accepted as is.

Reviewer #3 (Remarks to the Author):

The authors have correctly addressed all my suggestions for improvement.

I think that the work could still benefit from a minor clarification to the following point in the rebuttal letter:

"The radius of gyrations (Rgyr) and the collision cross sections (CCSs) in Figure S13 were calculated from the last frames of the ten simulations. And the details were added in the Methods on Page 11 and the legend of Figure S13."

Maybe I missed it, but I could not find the precise explanation regarding the "last frames" used for all the analysis in page 11 as stated above (Rgyr and CCS, now Figure S12). What does last frames mean? That should be clear to make the analysis easily reproducible by anyone in the future.

Response to Reviewers' comments:

We appreciate the positive tone and made every effort to address the issues raised in the reviews as detailed below:

Reviewer #3 (Remarks to the Author):

I think that the work could still benefit from a minor clarification to the following point in the rebuttal letter:

"The radius of gyration (R_{gyr}) and the collision cross sections (CCSs) in Figure S13 were calculated from the last frames of the ten simulations. And the details were added in the Methods on Page 11 and the legend of Figure S13."

Maybe I missed it, but I could not find the precise explanation regarding the "last frames" used for all the analysis in page 11 as stated above (R_{gyr} and CCS, now Figure S12). What does last frames mean? That should be clear to make the analysis easily reproducible by anyone in the future.

Response: The last frame means the conformation at the simulation time of 1200 ns, 450 ns, and 150 ns in the simulations at different temperatures: 300 K, 370 K and 370→450 K respectively. This information was added in the Methods on Page 11-12.